# AdipoRon Treatment Induces a Dose-Dependent Response in Adult Hippocampal Neurogenesis

**DOI:** 10.3390/ijms22042068

**Published:** 2021-02-19

**Authors:** Thomas H. Lee, Brian R. Christie, Henriette van Praag, Kangguang Lin, Parco Ming-Fai Siu, Aimin Xu, Kwok-Fai So, Suk-yu Yau

**Affiliations:** 1Department of Rehabilitation Sciences, Faculty of Health and Social Sciences, The Hong Kong Polytechnic University, Hong Kong; thomas.hy.lee@connect.polyu.hk; 2Division of Biomedical Sciences, University of Victoria, Victoria, BC V8P 5C2, Canada; brain64@uvic.ca; 3FAU Brain Institute and Charles E. Schmidt College of Medicine, Florida Atlantic University, Jupiter, FL 33431, USA; hvanpraag@health.fau.edu; 4Department of Affective Disorder, Guangzhou Brain Hospital, The Brain Affiliated Hospital of Guangzhou Medical University, Guangzhou 510370, China; linkangguang@163.com; 5Division of Kinesiology, School of Public Health, The University of Hong Kong, Hong Kong; pmsiu@hku.hk; 6Department of Medicine, The University of Hong Kong, Hong Kong; amxu@hku.hk; 7Department of Pharmacology and Pharmacy, The University of Hong Kong, Hong Kong; 8The State Key Laboratory of Pharmacology, The University of Hong Kong, Hong Kong; 9Guangdong-Hong Kong-Macau Institute of CNS Regeneration, Jinan University, Guangzhou 510632, China; hrmaskf@hku.hk; 10State Key Laboratory of Brain and Cognitive Sciences, The University of Hong Kong, Hong Kong; 11Department of Ophthalmology, The University of Hong Kong, Hong Kong

**Keywords:** AdipoRon, hippocampal neurogenesis, learning and memory, adiponectin, brain-derived neurotrophic factor

## Abstract

AdipoRon, an adiponectin receptor agonist, elicits similar antidiabetic, anti-atherogenic, and anti-inflammatory effects on mouse models as adiponectin does. Since AdipoRon can cross the blood-brain barrier, its chronic effects on regulating hippocampal function are yet to be examined. This study investigated whether AdipoRon treatment promotes hippocampal neurogenesis and spatial recognition memory in a dose-dependent manner. Adolescent male C57BL/6J mice received continuous treatment of either 20 mg/kg (low dose) or 50 mg/kg (high dose) AdipoRon or vehicle intraperitoneally for 14 days, followed by the open field test to examine anxiety and locomotor activity, and the Y maze test to examine hippocampal-dependent spatial recognition memory. Immunopositive cell markers of neural progenitor cells, immature neurons, and newborn cells in the hippocampal dentate gyrus were quantified. Immunosorbent assays were used to measure the serum levels of factors that can regulate hippocampal neurogenesis, including adiponectin, brain-derived neurotrophic factor (BDNF), and corticosterone. Our results showed that 20 mg/kg AdipoRon treatment significantly promoted hippocampal cell proliferation and increased serum levels of adiponectin and BDNF, though there were no effects on spatial recognition memory and locomotor activity. On the contrary, 50 mg/kg AdipoRon treatment impaired spatial recognition memory, suppressed cell proliferation, neuronal differentiation, and cell survival associated with reduced serum levels of BDNF and adiponectin. The results suggest that a low-dose AdipoRon treatment promotes hippocampal cell proliferation, while a high-dose AdipoRon treatment is detrimental to the hippocampus function.

## 1. Introduction

Adiponectin is the most abundant adipokine secreted by the adipose tissue [1]. It circulates as a trimer, hexamer, and multimer, but only trimeric and hexameric forms are detectable in human cerebrospinal fluid [2,3]. Rodent studies show that adiponectin can cross the blood-brain barrier [4,5]. Adiponectin levels in the central nervous system are approximately 1000 times lower than those in the circulation [2]. Human subjects with dementia [6,7,8] and depressive disorder [9,10] have significantly lower adiponectin levels in the circulation and the cerebrospinal fluid (CSF).

Adiponectin exerts its effects by binding to adiponectin receptor 1, receptor 2, or T-cadherin [11]. Both receptors 1 and 2 are expressed in the hippocampus [12], in which their levels are downregulated in Alzheimer’s disease (AD) patients [8]. Adiponectin plays a role in regulating hippocampal plasticity, including adult neurogenesis [13], dendritic arborization [14], and synaptic plasticity [15,16]. In 8–12 weeks old mice, adiponectin haploinsufficiency increases susceptibility to stress [13] while adiponectin deficiency impairs fear extinction [17]. Chronic adiponectin deficiency also impairs learning and causes memory deficits in older mice. In 12 months old mice, adiponectin deficiency impairs learning and memory performance in fear conditioning, object recognition, and Y-maze tests [16,18]. Up to 18 months old, these mice further display spatial memory impairment in the water maze task [18]. On the other hand, activating adiponectin signaling in the brain improves depressive behaviors [5,13] and hippocampal-dependent learning and memory deficits [16,17,19], in which hippocampal plasticity is involved. These findings collectively suggest that upregulating adiponectin expression or activating adiponectin signaling could be an effective strategy to improve hippocampal function by enhancing hippocampal neuroplasticity. 

Hippocampal plasticity is strongly linked to the metabolic condition of adipose tissue [20,21]. Adolescence is a critical window for shaping both adipose tissue properties and hippocampal plasticity. The proliferation of adipocyte is observed from childhood to adolescence with higher rates in obese individuals [22]. The adipocyte number remains constant in adulthood, but higher in obese adults [23]. Towards brain health, the adolescent brain is more vulnerable to metabolic stimuli than the adult brain, such as high-fat diet [24,25] and voluntary running exercise [26,27]. 

Circulating adiponectin levels are decreased in individuals with obesity and Type 2 diabetes [28]. Adiponectin possesses antidiabetic property [29,30,31] by promoting fatty acid oxidation through AdipoR1/AMPK or AdipoR2/PPAR-α pathway [32]. The antidiabetic effect of adiponectin can be mimicked by administering a small, orally active, adiponectin receptor agonist, AdipoRon [33]. Mechanistically, AdipoRon binds to both AdipoR1 and AdipoR2 in vitro with a greater affinity for AdipoR1 [33]. Functionally, the antidiabetic effect of AdipoRon is completely abolished under the condition of double knockout of AdipoR1 and AdipoR2, but not single knockout of either adiponectin receptors [33]. Further investigations reveal its anti-inflammatory [34], anti-atherosclerotic [35], hepatoprotective [36], and cardioprotective [37] properties in animal models. AdipoRon can be detected in the brain upon systemic injection [38] and oral administration [8], implicating its ability to cross the blood-brain barrier (BBB). Former studies focused on examining the pharmacological effect of AdipoRon in neurocognitive diseases [8,38], its neurotrophic effect on physiological condition has yet to be examined.

Our previous studies have shown that adiponectin mediates exercise-induced adult hippocampal neurogenesis in an AdipoR1/AMPK-dependent manner [5,39,40]. It is of our interest to investigate the potential effect of AdipoRon on adult hippocampal neurogenesis and the subsequent behavioral changes. Therefore, we examined whether chronic treatment with AdipoRon can affect hippocampal-dependent spatial memory and anxiety-like behavior, as well as hippocampal neurogenesis in a dose-dependent manner, and whether AdipoRon treatment modulates blood levels of neurogenesis-related factors, including brain-derived neurotrophic factor (BDNF), adiponectin, and corticosterone. 

## 2. Results

### 2.1. Chronic Treatment with 50 mg/kg AdipoRon Impaired Hippocampal-Dependent Spatial Memory 

One-way ANOVA revealed that chronic AdipoRon treatments of both dosages did not affect total traveling distance (Figure 1A; F_2,18_ = 0.468, *p* > 0.05) and mean speed (Figure 1B; F_2,18_ = 1.508, *p* > 0.05) in the open field test, indicating that AdipoRon did not affect locomotor activity. 

Both AdipoRon dosages did not affect anxiety-like behaviors, as shown by no significant difference in time spent at the center (Figure 1C; F_2,18_ = 1.874, *p* > 0.05), wall-hugging behaviors (Figure 1D; F_2,18_ = 1.557, *p* > 0.05), and defecation (Figure 1E; F_2,18_ = 1.708, *p* > 0.05).

Hippocampal-dependent spatial memory was examined by Y maze. 20 mg/kg AdipoRon treatment did not impair spatial recognition memory (Figure 1F; *p* > 0.05 vs. Vehicle), but 50 mg/kg treatment impaired spatial memory as indicated by a significant decrease in exploration index (Figure 1F; *p* < 0.005 vs. Vehicle). The data suggested that AdipoRon treatment at a high dose could impair spatial recognition memory. 

### 2.2. Treatment with 20 mg/kg AdipoRon Promoted Cell Proliferation, but 50 mg/kg AdipoRon Treatment Impaired Hippocampal Neurogenesis in the Dentate Gyrus

Hippocampal neurogenesis is involved in spatial learning and memory [41]. One-way ANOVA showed a significant effect of AdipoRon treatment on hippocampal cell proliferation (Figure 2A; F_2,12_ = 25.49, *p* < 0.0001). Post-hoc analysis showed that 20 mg/kg AdipoRon treatment significantly increased the number of proliferating cells (Figure 2A–C; *p* < 0.005 vs. Vehicle), whereas 50 mg/kg AdipoRon treatment suppressed cell proliferation (Figure 2A,D; *p* < 0.05 vs. Vehicle), highlighting differential effects of chronic treatment AdipoRon on cell proliferation.

AdipoRon treatment showed significant effects on the number of immature neurons labeled with doublecortin (DCX) in the dentate region (Figure 3A–D; F_2,12_ = 6.235, *p* < 0.05). Treatment with 20 mg/kg AdipoRon did not affect the number of immature neurons (Figure 3A; *p* > 0.05 vs. Vehicle), but 50 mg/kg treatment significantly reduced the number of immature neurons (Figure 3A; *p* < 0.05 vs. Vehicle). 

AdipoRon treatment also significantly modulated cell survival in new-born cells labeled with BrdU (Figure 4A–D; F_2,12_ = 25.36, *p* < 0.0001). Post-hoc analysis revealed that 20 mg/kg AdipoRon treatment did not affect new-born cell survival (Figure 4A; *p* > 0.05 vs. Vehicle), whereas 50 mg/kg AdipoRon treatment significantly reduced cell survival, as evidenced by a significant decrease in BrdU positive cells (Figure 4A; *p* < 0.005 vs. Vehicle). 

Lastly, quantification of BrdU/DCX co-labeled cells showed that AdipoRon treatment significantly affected neuronal differentiation (Figure 4E–H; F_2,12_ = 9.56, *p* < 0.005). Post-hoc test revealed that 20 mg/kg AdipoRon, but not 50 mg/kg AdipoRon treatment, significantly suppressed neuronal differentiation (Figure 4E; *p* < 0.05 vs. Vehicle). 

### 2.3. Chronic Treatment with AdipoRon Affects Serum Levels of BDNF, Adiponectin, and Corticosterone

Chronic treatment with AdipoRon significantly modulated serum levels of BDNF (Figure 5A; F_2,12_ = 93.83, *p* < 0.0001). Treatment with 20 mg/kg AdipoRon significantly increased serum BDNF levels (*p* < 0.005 vs. Vehicle), which was reduced by 50 mg/kg AdipoRon (Figure 5A; *p* < 0.005 vs. Vehicle). Similar effect was also observed in serum adiponectin (Figure 5B; F_2,12_ = 70.25, *p* < 0.0001). Treatment with 20 mg/kg AdipoRon significantly increased serum adiponectin levels (Figure 5B; *p* < 0.005 vs. Vehicle), whereas 50 mg/kg AdipoRon treatment reduced it (Figure 5B; *p* < 0.005 vs. Vehicle). Glucocorticoids suppresses neurogenesis [42], interestingly, the results showed that treatment with AdipoRon increased serum corticosterone levels regardless of the treatment dosages (Figure 5C; F_2,12_ = 8.909, *p* < 0.005). Of note, blood glucose levels were unaffected by AdipoRon treatment in all groups (Figure 5D; F_2,12_ = 1.705, *p* > 0.05 vs. Vehicle). 

## 3. Discussion

The present study showed that chronic AdipoRon treatment at a low dose (20 mg/kg) and a high dose (50 mg/kg) have differential effects on hippocampal-dependent spatial learning, hippocampal neurogenesis, and the levels of serum biomarkers that can modulate neurogenesis. A low-dose AdipoRon treatment promoted cell proliferation in the hippocampal dentate gyrus with increased serum adiponectin and BDNF levels. Conversely, a high-dose treatment impaired spatial memory and suppressed hippocampal neurogenesis with reduced serum adiponectin and BDNF levels. The findings collectively suggested that chronic AdipoRon treatment at a high dose could be detrimental to hippocampal functions by impairing hippocampal neurogenesis (Figure 6). 

Cell quantification showed that a low-dose (20 mg/kg) AdipoRon treatment promoted hippocampal cell proliferation with increased serum levels of adiponectin and BDNF. With the ability to cross the blood-brain barrier [5], adiponectin may mediate the AdipoRon action on hippocampal cell proliferation. The direct action of adiponectin on hippocampal cell proliferation is evidenced in vivo [14] and in vitro [5,43]. Besides, BDNF mediates hippocampal cell proliferation through TrkB [44]. Circulating and brain BDNF levels are positively correlated in humans and rodents [45]. The elevated serum BDNF level echoes our latest finding, illustrating that a low-dose AdipoRon treatment raises BDNF level in the dentate subregion (Lee et al., unpublished data). While the action of BDNF encompasses all stages of adult neurogenesis [46], we did not observe any neurotrophic effects of AdipoRon on neuronal differentiation and survival at a low dose. We speculate that BDNF could play a more critical role in forming dendritic arbors and spines [47]. A recent study shows that AdipoRon administration promotes spine formation in the apical dendrites of CA1 neurons in both 5 × FAD and 5 × FAD, *Adipo*-knockout mice [8], while we also observed an increased spine density in DG granule neurons in mice receiving low-dose AdipoRon treatment (Lee et al., unpublished data). The ability to cross BBB facilitates the direct action of AdipoRon on the hippocampus [8,38], where adiponectin receptors are expressed [12]. Intracerebral ventricular infusion of AdipoRon promotes cell proliferation in DG of APP/PS1 mice, and AdipoRon also restores Aβ-impaired neural stem cell proliferation [48]. Collectively, this study presents the potential direct and indirect actions of AdipoRon on hippocampal cell proliferation.

Meanwhile, the specific roles of AdipoR1 and AdipoR2 in adult hippocampal neurogenesis remain elusive. Full-length and dodecameric (high molecular weight, HMW) adiponectin have higher affinities for AdipoR2, while trimeric and globular adiponectin have higher affinities for AdipoR1 [49]. Trimeric adiponectin can be found in human CSF [2]. Our team has reported that trimeric adiponectin promotes neural progenitor cell proliferation in vitro in an AdipoR1-dependent manner by siRNA knockdown. Another study shows that both globular and full-length adiponectin promote neural progenitor cell proliferation in vitro [43]. The direct effect of full-length adiponectin suggests the potential involvement of AdipoR2 in modulating cell proliferation. Moreover, AdipoR2 may play a critical role in dendritic spine formation and granule neuron excitability in the dentate gyrus. Electrophysiological study reveals that AdipoR2 and adiponectin knockout induces hyperexcitability of the dentate granule neurons [17]. Chronic neuronal hyperexcitability could contribute to dendritic spine degeneration [50,51,52,53,54,55,56,57]. Notably, the spine density of dentate granule neurons is reduced in adiponectin deficiency [14]. Activating the adiponectin signaling by AdipoRon and adiponectin can restore hyperexcitability in dentate granule neurons [17] and promote dendritic spine formation [14], respectively. These findings warrant further investigation to examine whether AdipoR2 mediates dendritic spine formation in the dentate granule neurons.

In this study, AdipoRon treatment began in adolescence from six weeks of age. We cannot rule out the possibility that this development stage may play a role in affecting the proliferative effect of AdipoRon. The hippocampal cell proliferation rate is higher in adolescence than the adult and aged mice [58]. Besides, hippocampal cell proliferation is more responsive to drug treatment, like fluoxetine, in juvenile than in young adult brains [59]. 

In contrast, 50 mg/kg AdipoRon treatment is regarded as a high-dose treatment since serum adiponectin level was significantly reduced, implicating a homeostatic change by reducing adiponectin synthesis from the adipose tissues [60]. This adaptive response in adipose tissue is reported when adipocyte differentiation is suppressed by prolonged AdipoRon treatment (20 μM, 8 days) in C3H10T1/2 mouse embryonic mesenchymal stem cells [60]. The anti-adipogenic effect of AdipoRon is associated with reduced transcript expressions of adipocyte-specific markers, including adiponectin, but increased AMPK and ACC phosphorylation. Thus, a high-dose AdipoRon treatment may initiate negative feedback by reducing adiponectin secretion and suppressing adipogenesis to overcome hyperactivation of adiponectin receptor-mediated cascades. 

AMPK is a downstream target of adiponectin [61] and AdipoRon [33]. AMPK is required to facilitate AdipoRon-induced hippocampal cell proliferation in APP/PS1 mice and in vitro, whereas Compound C, an AMPK antagonist, blunts the AdipoRon action [48]. We speculate the high-dose treatment may hyperactivate the AMPK signaling in the hippocampus. Overactivation of AMPK by AICAR infusion or AMPK overexpression induces AMPK^Thr172^ phosphorylation but reduces BDNF expression in the hippocampal CA1 [62]. Memory deficit is subsequently presented in contextual fear conditioning after AMPK activation [62]. The abovementioned investigation only represents the scenario of AMPK hyperactivation. It is noteworthy that such pro-cognitive treatments as voluntary wheel running [5] also promotes AMPK activation in the hippocampus. The role of AMPK activation serves as an adaptive response of metabolic reprogramming [63]. Likewise, low-dose AdipoRon treatment increased AMPK^Thr172^ phosphorylation in the dentate subregion (Lee et al., unpublished data). 

AMPK activation can inhibit mTOR [64], whereas activating mTOR pathway promotes BDNF expression in the hippocampus [65]. Reduced BDNF levels impair neuronal differentiation and survival in the hippocampus [66], while depleting BDNF in the hippocampus impairs hippocampal-dependent spatial learning tasks [67]. Consistently, our results showed that a high-dose AdipoRon treatment abrogated hippocampal neurogenesis in concurrent with reductions in serum BDNF levels. Significant impairment in adult hippocampal neurogenesis may contribute to learning and memory deficits [68]. Collectively, the high-dose AdipoRon treatment may hyper-activate AMPK signaling, which in turn, inhibits mTOR activity and reduces BDNF levels, and consequently impairs hippocampal neurogenesis under physiological condition.

Lastly, AdipoRon increases circulating corticosterone levels upon treatment. The underlying reason could be explained by the activation of the adiponectin receptors in the adrenal gland. Adiponectin treatment increases corticosterone secretion in rat adrenocortical cells in vitro in a dose-dependent manner [69]. Caloric restriction [70] and physical exercise [71] are mild stressors that increase corticosterone levels, but they are effective in increasing BDNF levels and promote adult hippocampal neurogenesis [72]. Physical exercise could promote brain BDNF in rodents [73] and circulating BDNF in humans [74]. Caloric restriction [75] and physical exercise [76] can also enhance adiponectin levels. Hence, the increased BDNF and adiponectin levels may be sufficient to overcome the suppressive effects of corticosterone on hippocampal neurogenesis as physical exercise does [77]. 

## 4. Materials and Methods

### 4.1. Animals 

Five-week-old male wild-type mice with C57BL/6J genetic background were housed in the Centralized Animal Facilities at The Hong Kong Polytechnic University. The animals were fed with standard chow and water ad libitum and kept under a 12-h:12-h light-dark cycle. All experimental procedures were approved and followed the Animal Subjects Ethics Sub-Committee’s guidelines, The Hong Kong Polytechnic University.

### 4.2. Drug Preparation and Treatment

AdipoRon (Sigma-Aldrich, MO, USA) was first dissolved in DMSO at 40 mg/mL, then suspended in 0.5% carboxymethylcellulose salt (Sigma-Aldrich, St. Louis, MO, USA) [35,78]. Animals received the AdipoRon treatment (20 or 50 mg/kg i.p.) or the vehicle as control treatment continuously for 14 days. Mice were injected with bromodeoxyuridine i.p. (BrdU: 50 mg/kg dissolved in 0.9% saline) before AdipoRon treatment to study newborn cell survival. The day after the last treatment, mice were subjected to an open field test and Y-maze task, as illustrated in Figure 7.

### 4.3. Open Field Test

Mice were first brought to the testing room for 2-hr adaptation. Each mouse was allowed to explore the open field (L × W × H: 40 × 40 × 30 cm^3^) for 10 min. Between trials, open field was cleaned by 70% ethanol wipe and two rounds of water wipes. The test box was dried before the next trial. Anxiety-like behavior and locomotor activity were analyzed using ANYMAZE software (Stoelting Co., Wood Dale, IL, USA). Locomotor activity was calculated as the total distance traveled over 10 min. Thigmotaxis is linked to anxiety-like behaviors [79,80,81]. Wall-hugging is counted when the animal performed vertical rear against the wall supported by two hind paws [82]. Wall sniffing behavior was excluded. Defecation is another indicator of anxiety [83]. The number of fecal boli was counted.

### 4.4. Y-Maze Task

Spatial recognition memory was assessed by the Y-maze task [84]. Three identical arms (L × W × H: 10 × 6 × 8 cm^3^) in Y-maze were pre-designated as starting arm, familiar arm, and novel arm, respectively. Three arms were counterbalanced to avoid recognition bias. In the 10-min habituation session, a mouse was released in the designated starting arm and was allowed to explore the maze with a blocked novel arm. After a 4-hr intermission time, the animal re-explored the maze with the unblocked novel arm for 5 min. Time spent in each arm was hand-scored by the trained researcher in a sample blinded manner. The exploration index was calculated as (N − F)/(N + F) and presented on a −1 to 1 scale.

### 4.5. Tissue Preparation

Mice were deeply anaesthetized with isoflurane (Zoetis, Surrey, UK). Blood was transcardially collected by a 25-G needle with a 1 mL syringe. Samples were clotted for 30 min at room temperature. Blood sera were then collected by centrifugation at 1000× *g* for 20 min at 4 °C and then stored at −80 °C until ELISA measurements. Animals were rapidly perfused with 0.9% saline, followed by fixation with 4% paraformaldehyde (PFA) in 0.01 M phosphate-buffered saline (PBS). The isolated brains were post-fixed in 4% PFA at 4 °C overnight. The brains were then transferred to 30% sucrose until they sank. Coronal sections (1-in-6 series, 30-μm thickness) were obtained using a vibratome (Leica Biosystems, Heidelberg, Germany). The slices were temporarily stored in a cryoprotectant composed of 30% glycerol and 30% ethylene glycol at 4 °C until immunostaining. 

### 4.6. Immunohistochemistry and Immunofluorescent Staining

Immunostaining was performed by the free-floating method, as previously performed [85]. For BrdU staining, the antigens were retrieved in the citric acid buffer (pH 6.0) at 95 °C for 15 min, then denatured in 2 N HCl for 30 min and neutralized by 0.1 M borate buffer (pH 8.5) for 15 min at room temperature. The sections were incubated overnight with mouse anti-BrdU antibody (1:1000; Roche Life Science, Upper Bavaria, Germany), and then incubated with the biotinylated goat anti-mouse IgG (1:200; Vector Laboratories, Burlingame, CA, USA) for 2 h at room temperature. BrdU staining was visualized with the peroxidase method using the VECTASTAIN^®^ ABC kit (HRP) (Vector Laboratories, Burlingame, CA, USA) and the DAB peroxidase substrate kit (Vector Laboratories, Burlingame, CA, USA). For doublecortin (DCX) and Ki-67 staining, sections were incubated with mouse anti-DCX (1:200; Santa Cruz Biotechnology, Dallas, TX, USA) or rabbit anti-Ki67 (1:1000; abcam, Cambridge, UK) antibody, then secondary antibodies, the biotinylated goat anti-mouse, or goat anti-rabbit IgG (1:200; Vector Laboratories, Burlingame, CA, USA), and visualized using the peroxidase method.

Immunofluorescent co-labeling of BrdU and DCX was performed as previously described [5,86]. After antigen retrieval, sections were incubated with primary antibodies overnight and secondary antibodies for 2 h, including goat anti-rabbit IgG Alexa Fluor-488 and goat anti-mouse IgG Alexa Fluor-568. The mounted sections were coverslipped with the fluorescent mounting medium (Dako, Carpinteria, CA, USA). 

### 4.7. Quantification of BrdU, Ki67, and DCX Immunopositive Cells

BrdU-, Ki67-, and DCX-positive cells were counted in the 1-in-6 series of nine sections (from bregma −1.34 to −3.80 mm), using an optical fractionator system (grid size: 55 μm × 55 μm, counting frame: 35 μm × 35 μm) of StereoInvestigator (MicroBrightfield Inc., Williston, VT, USA) as previously described [5]. Cells located in the DG sub-granular zone and granular cell layer were counted, whereas those located in the uppermost focal plane were excluded. Quantification was performed in a sample-blinded manner.

### 4.8. Quantification of DCX/BrdU Co-Labeled Cells

Images of six sections from each animal were captured using LSM 800 confocal laser scanning microscope (Carl Zeiss Microscopy, White, Plains, NY, USA). Fifty BrdU positive cells were randomly selected to confirm the BrdU/DCX co-labeled ratio as an indicator for neuronal differentiation. Quantification of the co-labeling was performed in a sample blinded manner. 

### 4.9. Measurement of Serum Glucose Levels, Corticosterone, Adiponectin and Brain-Derived Neurotrophic Factor (BDNF) Levels

Blood glucose from tail blood was measured using the Accu-Chek© Performa glucometer (Roche, Sydney, Australia) before sacrificing the animals. Serum levels of adiponectin, BDNF, and corticosterone were determined by commercial ELISA kits, including corticosterone ELISA kit (Assay Designs, Enzo Life Sciences, Lausen, Switzerland), mouse adiponectin ELISA kit (Immunodiagnostics, Hong Kong, China), and the total BDNF Quantikine ELISA kit (R & D system, Minneapolis, MN, USA) according to the manufacturer’s instructions.

### 4.10. Statistical Analyses

One-way ANOVA with Tukey’s posthoc test was performed using GraphPad Prism version 8.0.0 for Windows (GraphPad Software, San Diego, CA, USA). A probability (P) value of less than 0.05 is considered statistically significant. Data were shown as mean ± SEM.

## 5. Conclusions

In summary, our data demonstrated that a low-dose AdipoRon treatment promoted hippocampal cell proliferation and increased serum adiponectin and BDNF levels. In contrast, a high-dose of AdipoRon treatment was deleterious to hippocampal function, as evidenced by learning and memory deficits in hippocampus-dependent tasks. The ability to cross the BBB may allow AdipoRon to promote cell proliferation directly, though a low-dose AdipoRon treatment increased BDNF levels to elicit its neurotrophic effect indirectly. On the other hand, reduced neurotrophic factors and aberrant hippocampal neurogenesis might contribute to spatial memory with a high-dose AdipoRon treatment. The present study provides evidence of the beneficial effects of AdipoRon treatment on inducing hippocampal cell proliferation and the potential side effects of a high dose AdipoRon treatment on hippocampal neurogenesis impairment with learning and memory deficits.

## Figures and Tables

**Figure 1 ijms-22-02068-f001:**
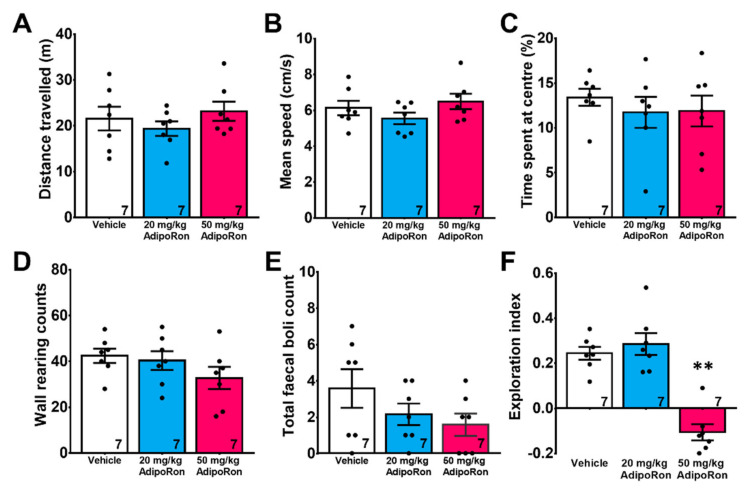
Effects of AdipoRon on rodent behaviors. Administration of AdipoRon at a low dose (20 mg/kg) and high dose (50 mg/kg) for two weeks did not affect (**A**) locomotor activity and (**B**) mean speed traveled in the open field. Both dosages did not affect anxiety-like behaviors as indicated by (**C**) the percentage time spent at the center, (**D**) the number of vertical rears, and (**E**) the number of fecal boli in the open field. However, treatment with 50 mg/kg AdipoRon (high dose) impaired spatial recognition memory, as indicated by (**F**) reduced exploration index in the Y-maze task. The results were expressed as the mean ± SEM. *n* = 7 per group, ** *p* < 0.005 compared to the vehicle treated control.

**Figure 2 ijms-22-02068-f002:**
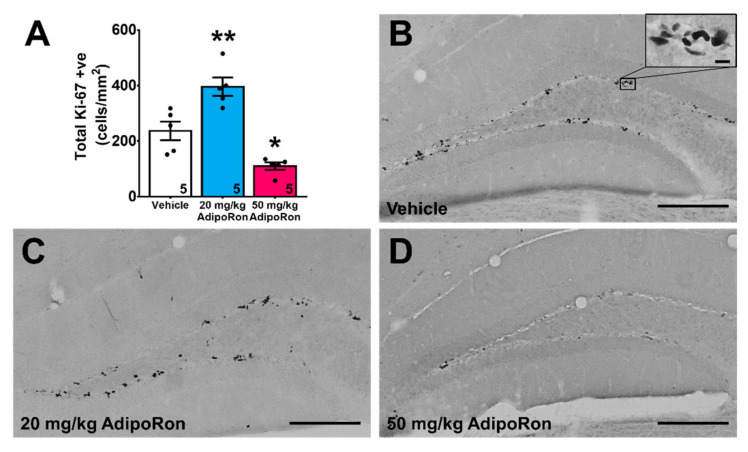
Effects of AdipoRon treatment on neural progenitor cell density. (**A**) Treatment with 20 mg/kg AdipoRon increased Ki67 positive cells. A total of eight sections were quantified in each sample. The results for the data were expressed as the mean ± SEM. *n* = 5 per group, * *p* < 0.05, ** *p* < 0.005 compared to the vehicle treated control. Representative image of Ki67 positive cells in the (**B**) control, (**C**) 20 mg/kg AdipoRon, and (**D**) 50 mg/kg AdipoRon treatment groups (Scale bars, 200 μm in 100×, 10 μm in 400×).

**Figure 3 ijms-22-02068-f003:**
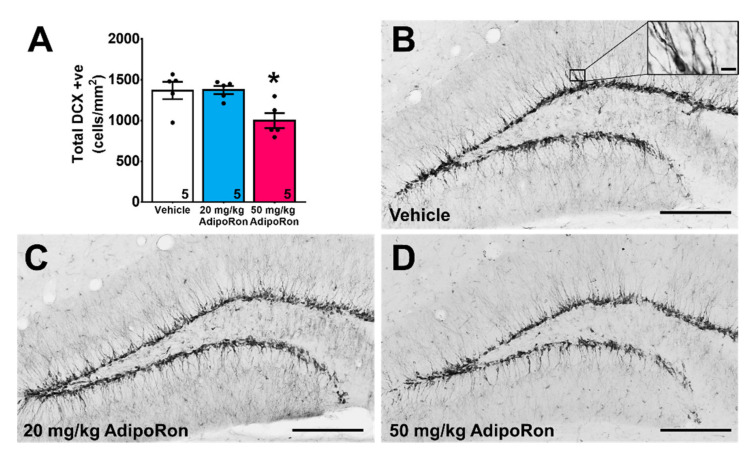
Effects of AdipoRon treatment on the immature neuron density. (**A**) Treatment with 50 mg/kg AdipoRon significantly decreased DCX positive cells. A total of eight sections were quantified in each sample. The results were expressed as the mean ± SEM. *n* = 5 per group, * *p* < 0.05 compared to the control. Representative images of DCX-positive cells in (**B**) control animal, mice receiving (**C**) 20 mg/kg, and (**D**) 50 mg/kg AdipoRon treatments (Scale bars, 200 μm in 100×, 10 μm in 400×).

**Figure 4 ijms-22-02068-f004:**
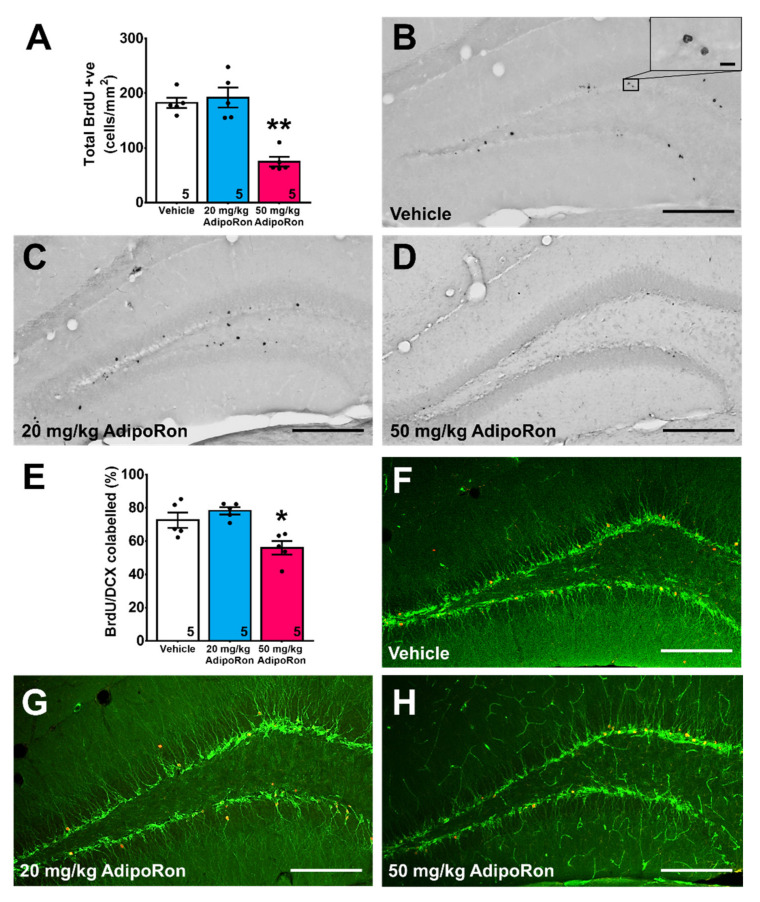
Effects of AdipoRon treatment on newborn cell density and neuronal differentiation. (**A**) Treatment with 50 mg/kg AdipoRon significantly decreased cell survival. A total of eight sections were quantified in each sample. The results were expressed as the mean ± SEM. *n* = 5 per group, ** *p* < 0.005 compared to the vehicle-treated control. Representative image of BrdU positive cells in (**B**) control animal, (**C**) 20 mg/kg, and (**D**) 50 mg/kg AdipoRon treatment groups. (**E**) Treatment with 50 mg/kg AdipoRon significantly decreased neuronal differentiation in the dentate as estimated by the co-labeling ratio of BrdU and DCX positive cells. A total of 50 BrdU positive were selected from six sections. The results for the data were expressed as the mean ±SEM. *n* = 5 per group, * *p* < 0.05 compared to the vehicle-treated control. Representative image of BrdU/DCX-labeled cells in (**F**) control animal, (**G**) 20 mg/kg, and (**H**) 50 mg/kg AdipoRon treatment groups. * *p* < 0.05, ** *p* < 0.005 compared to the vehicle-treated control. (Scale bars, 200 μm in 100×, 10 μm in 400×).

**Figure 5 ijms-22-02068-f005:**
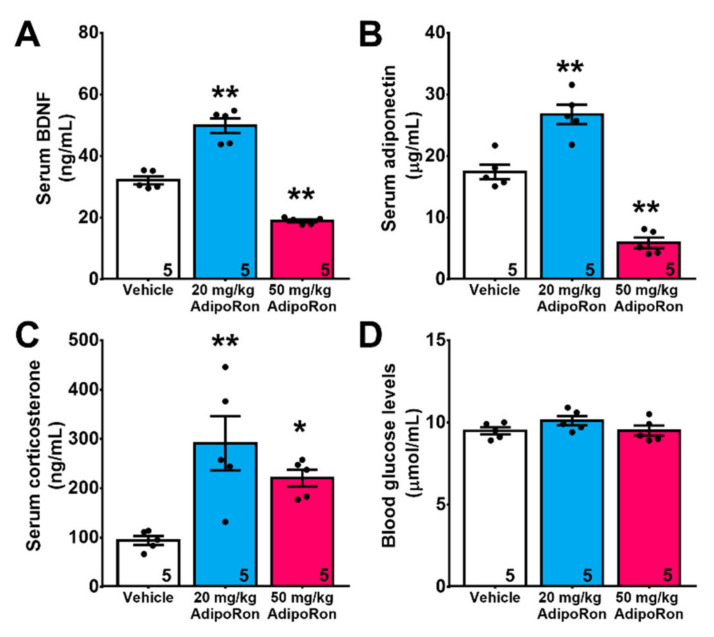
Effect of AdipoRon on blood levels of adiponectin, BDNF, corticosterone, and blood glucose levels. Treatment with 20 mg/kg AdipoRon levels significantly increased blood levels of (**A**) BDNF and (**B**) adiponectin levels, whereas treatment with 50 mg/kg AdipoRon showed opposite effects. (**C**) AdipoRon treatment increased serum levels of corticosterone but showed no effect on (**D**) blood glucose levels. The results were expressed as the mean ± SEM. *n* = 5 per group, * *p* < 0.05, ** *p* < 0.005 compared to the vehicle-treated control.

**Figure 6 ijms-22-02068-f006:**
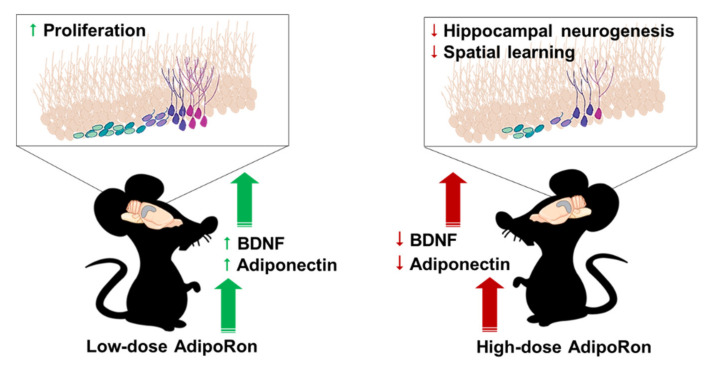
Summary of AdipoRon actions on hippocampal function. A low-dose AdipoRon treatment significantly promoted hippocampal cell proliferation and increased serum levels of adiponectin and BDNF. Conversely, a high-dose AdipoRon treatment suppressed hippocampal neurogenesis, reduced serum levels of adiponectin and BDNF, leading to hippocampal-dependent memory deficits. Created with biorender.com.

**Figure 7 ijms-22-02068-f007:**
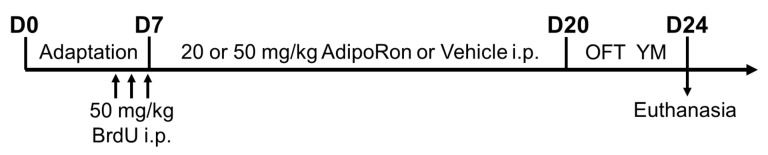
Timeline of AdipoRon treatment and behavioral tests. Before AdipoRon treatment, mice received three doses of BrdU injections (50 mg/kg i.p.) to study newborn cell survival. Mice have received AdipoRon treatment (20 or 50 mg/kg i.p.) or vehicle treatment continuously for 14 days. The day after, mice were subjected to the open field test and the Y-maze task on two consecutive days. The day after the Y-maze test, mice were sacrificed to collect blood samples for ELISA and fixed brain tissues for immunostaining.

## Data Availability

The data that support the findings of this study are available from the corresponding author upon reasonable request.

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
