# Peer review of "AdipoRon Treatment Induces a Dose-Dependent Response in Adult Hippocampal Neurogenesis"

_ijms, 2021, doi:10.3390/ijms22042068_

Round 1

Reviewer 1 Report

Major comments:

  1. In the Introduction, the authors claimed that "adiponectin-deficient mice display learning and memory deficits in Morris water maze and fear conditioning tasks (14). " The authors did not mention that this study was performed in aged male mice to investigate the Alzheimer's disease-like cognitive impairments. In another study conducted in adult mice, it shows that adiponectin deficiency could result in impaired extinction of contextual fear, which means long-lasting fear memory [1]. The authors should give an unbiased opinion since this study was performed in young mice.
  2. Five-week-old male wild-type mice were used in this study. The mice were at their adolescence. Could the developmental factors play a role in the adipoRon’s effect? This paper studies the adult hippocampal neurogenesis, why the authors did not use adult mice?
  3. Why the authors chose 20 mg and 50 mg as low dose and high dose respectively, based on what studies?
  4. The authors calculated the percentage time spent in the center as the only parameter to investigate the anxiety-like behaviors. It has been proposed that measuring anxiety in rodent models is much more complicated than using a single parameter in a single maze environment [2]. Did the authors record thigmotaxis or wall-hugging behavior? How about the number of fecal pellets left in the “open field” after the test? Elevated plus maze is suggested to strengthen evaluation of the anxiogenic results.
  5. Did the authors clean the “open field” between each trials? There was no relative description in the method.
  6. What is the affinity of adipoRon to adipoR1 and adipoR2 receptors? Both AdipoR1 and AdipoR2 are highly expressed in the dentate gyrus [3]. The authors emphasized adipoR1/AMPK-dependent pathway for adipoRon’s effects on modulating neurogenesis. Is it possible that adipoR1 and adipoR2 have opposite effects for neurogenesis?

  1. https://www.nature.com/articles/mp201658
  2. https://pubmed.ncbi.nlm.nih.gov/18690100/
  3. https://www.pnas.org/content/109/30/12248

Author Response

We would like to thank you for your active participation in the review process. We have greatly improved our manuscript in this revision, and hope that the quality is beyond satisfactory. Please see the following point-to-point responses.

In the Introduction, the authors claimed that "adiponectin-deficient mice display learning and memory deficits in Morris water maze and fear conditioning tasks (14). " The authors did not mention that this study was performed in aged male mice to investigate the Alzheimer's disease-like cognitive impairments. In another study conducted in adult mice, it shows that adiponectin deficiency could result in impaired extinction of contextual fear, which means long-lasting fear memory [1]. The authors should give an unbiased opinion since this study was performed in young mice.

            Thank you for your suggestion. We have revised the introduction, emphasizing the age range where behavioural deficits appear in adiponectin deficient condition. Please see Introduction, Line 56-62:

“In 8-12 weeks old mice, adiponectin haploinsufficiency increase susceptibility to stress (13) while adiponectin deficiency impairs fear extinction (17). Chronic adiponectin deficiency also impairs learning and memory deficits in older mice. In 12 months old mice, adiponectin deficiency impairs learning and memory performance in fear conditioning, object recognition, and Y-maze tests (16, 18). Up to 18 months old, these mice further display spatial memory impairment in the water maze task (18).”

Five-week-old male wild-type mice were used in this study. The mice were at their adolescence. Could the developmental factors play a role in the adipoRon’s effect? This paper studies the adult hippocampal neurogenesis, why the authors did not use adult mice?

            Thank you for pointing this out. We have emphasized the potential age effects as shown in Introduction, Line 68-74 & Discussion, Line 252-256

“Hippocampal plasticity is strongly linked to the metabolic condition of adipose tissue (20, 21). Adolescence is a critical window for shaping both adipose tissue properties and hippocampal plasticity. The proliferation of adipocyte is observed from childhood to adolescence with higher rates in obese individuals (22). The adipocyte number remains constant in adulthood, but higher in obsess adults (23). Towards brain health, the adolescent brain is more vulnerable to metabolic stimuli than the adult brain, such as high-fat diet (24, 25) and voluntary running exercise (26, 27).”

“In this study, AdipoRon treatment began in adolescence from six weeks of age. We cannot rule out the possibility that this development stage may play a role in affecting the proliferative effect of AdipoRon. Hippocampal cell proliferation rate is higher in the adolescence than the adult and aged mice (58). Besides, hippocampal cell proliferation is more responsive to drug treatment like fluoxetine in juvenile than in young adult brains (59).”

Why the authors chose 20 mg and 50 mg as low dose and high dose respectively, based on what studies?

 We started the experiment with a daily dosage of 20 mg/kg AdipoRon based on Okada-Iwabu et al paper demonstrating that continuous AdipoRon administration (50 mg/kg, 10 days, oral administration) reduces insulin resistance, glycaemic index, and fatty acid metabolism under metabolically compromised conditions in high-fed diet-fed mice and db/db mice. We, therefore, started with two dosages to test the chronic treatment effect on cognitive performance.

The authors calculated the percentage time spent in the center as the only parameter to investigate the anxiety-like behaviors. It has been proposed that measuring anxiety in rodent models is much more complicated than using a single parameter in a single maze environment [2].

Did the authors record thigmotaxis or wall-hugging behavior? How about the number of fecal pellets left in the “open field” after the test?

Thank you for your suggestion. We have analyzed the number of times where the animal performed wall rearing activity and the faecal boli from our pre-recorded videos. The new figures were included. One-way ANOVA showed that AdipoRon treatment did not affect thigmotaxis and defecation. Tukey’s post-hoc test showed no significant difference among groups. Section 2.1, Line 105 - 108 and Figure 1D, 1E:

“Both AdipoRon dosages did not affect anxiety-like behaviours as shown by no significant difference in time spent at centre (Figure 1C; F2,18 = 1.874, P > 0.05), wall-hugging behaviours (Figure 1D; F2,18 = 1.557, P > 0.05), and defecation (Figure 1E; F2,18 = 1.708, P > 0.05).”

Elevated plus maze is suggested to strengthen evaluation of the anxiogenic results.

We have performed open field test, but not elevated plus maze to examine anxiety-like behaviour. We will take this advice to include elevated plus maze in the future study examining anxiety-like behaviour.

Did the authors clean the “open field” between each trials? There was no relative description in the method.

Yes. We have clarified our procedures in Methods and Materials, Line 335 - 337:

“Between trials, open field was cleaned by 70% ethanol wipe and two rounds of water wipes. The test box was dried before the next trial.”

What is the affinity of AdipoRon to adipoR1 and adipoR2 receptors?

AdipoRon has a stronger affinity for AdipoR1. We have addressed it in Introduction, Line 79 - 82:

“Mechanistically, AdipoRon binds to both AdipoR1 and AdipoR2 in vitro with a greater affinity for AdipoR1 (33). Functionally, the antidiabetic effect of AdipoRon is completely abolished under the condition of double knockout of AdipoR1 and AdipoR2, but not single knockout of either adiponectin receptors (33).”

Both AdipoR1 and AdipoR2 are highly expressed in the dentate gyrus [3]. The authors emphasized adipoR1/AMPK-dependent pathway for AdipoRon’s effects on modulating neurogenesis. Is it possible that adipoR1 and adipoR2 have opposite effects for neurogenesis?

Thank you for pointing this out. We have further addressed this in Discussion, Line 234-251:

“Meanwhile, the specific roles of AdipoR1 and AdipoR2 in adult hippocampal neurogenesis remain elusive. Full-length and dodecameric (high molecular weight, HMW) adiponectin have higher affinities for AdipoR2, while trimeric and globular adiponectin have higher affinities for AdipoR1 (49). Trimeric adiponectin can be found in human CSF (2). Our team has reported that trimeric adiponectin promotes neural progenitor cell proliferation in vitro in an AdipoR1-dependent manner by siRNA knockdown. Another study shows that both globular and full-length adiponectin promote neural progenitor cell proliferation in vitro (43). The direct effect of full-length adiponectin suggests the potential involvement of AdipoR2 in modulating cell proliferation. Moreover, AdipoR2 may play a critical role in dendritic spine formation and granule neuron excitability in the dentate gyrus. Electrophysiological study reveals that AdipoR2 and adiponectin knockout induces hyperexcitability of the dentate granule neurons (17). Chronic neuronal hyperexcitability could contribute to dendritic spine degeneration (50-57). Notably, the spine density of dentate granule neurons is reduced in adiponectin deficiency (14). Activating the adiponectin signalling by AdipoRon and adiponectin can restore hyperexcitability in dentate granule neurons (17) and promote dendritic spine formation (14), respectively. These findings warrant further investigation to examine whether AdipoR2 mediates dendritic spine formation in the dentate granule neurons.”

Reviewer 2 Report

The paper by Lee and colleagues is interesting, but I feel as though further experimental evidences are lacking in order to make the results more significant and relevant for the field. Indeed, it would be interesting to include data correlating the observed evidences with AMPK activation, BDNF direct increase or its effects on synaptogenesis.

Moreover, I have some suggestions I feel the authors should take into consideration before the manuscript can be accepted for publication:

- I would adjust the title as I don’t believe the part stating “but high dose shows opposite effects” fully depicts the meaning of the paper (maybe something along the lines of “AdipoRon treatment induces a dose-dependent response in adult hippocampal neurogenesis”)

- The authors need to address some English mistakes, for example:

         - Abstract: line 26 “yet to be examined”; line 28 “either 20 mg/kg (low dose) or 50 mg/kg (high dose)..”; line 37 “Conversely” is not correct as the 50 mg/kg dose shows the opposite effect, not a concordant one (so maybe “on the contrary?”): line 39 “the results suggest…”

         - Introduction: line 46 “secreted by the adipose tissue”

         - Results, line 124 I would rephrase as “highlighting differential effects of …”

         - Discussion, line 199 “the findings echoed previous reports..”; The sentence in line 199-202 is not clear.; line 208 “BDNF is a neurotrophic factor THAT promotes adult neurogenesis”

- In the introduction, it is not clear why adiponectin’s anti-diabetic effects are mentioned right away when discussing its effect on the brain, maybe they should be commented on when discussing the action of adipoRon in diabetes (e.g. line 69). Moreover, if there is more than 1 paper discussing the neuroprotective effects of AdipoRon (other than ref 20), they should be included and commented upon

- In the results, when describing Figure 1 A-C, it should be stated that the results refer to both dosages (none of them affects travelling distance/speed)

- In Figure 1 (and in the subsequent figures), if the number indicated in the column (7) refers to the number of animals analyzed, this should be stated in the legend. Moreover, it should be stated here if the data refers to the mean +/- SD or SEM.

- In figure 1 the legend is present only in Figure 1C. I understand that this is the same for the adjacent panels, but it is misleading that there is no clear labeling in Panel 1A and 1B, I would thus either add the legend on the X axis or anyways add it to make it clearer. The same applies for Figure 5

- I think Figure 1D1 is a bit redundant to Figure 1D2 and confusing, I suggest keeping only Figure 1D2 which gives a clearer idea of the result obtained

- In figure 2, how many images were stained and quantified per condition? This should be stated in the legend (I see it is present in the methods but it would be useful if it were reported here too). Moreover, scale bars are present only in the magnification of figure 2 and in Figure 2D, they should be present in all images (This is also true for Figure 3 and Figure 4)

- In line 153, the authors state that “20 mg/kg AdipoRon, but not 50 mg/kg AdipoRon treatment, significantly suppressed neuronal differentiation” but in Figure 4E the opposite seems to be true (The 50 dosage suppresses neuronal differentiation).

- Figures 4F-4H should be commented upon in the text. Moreover, it is not clear to what condition these imaged refer to: The vehicle, the 20 dosage or the 50 dosage. Representative images of the staining for each condition should be reported.

- The authors should elaborate on how they believe that “The increased adiponectin and BDNF levels may contribute to the indirect action of AdipoRon on adult neurogenesis.” (line 209)

- Do the authors have any evidence of hippocampal expression of BDNF? I see they state that evidences demonstrate a positive correlation between circulating and brain bdnf levels (208), but the actual BDNF concentration in the brain could be useful.

- The authors state that “BDNF could play a more critical role in dendritic morphogenesis (27). It will be of interest to examine dendritic structures of newborn neurons in future studies.” (213-214). Do they have any evidence of this, such as immunohistochemical staining or gene expression analysis? I believe this would enrich the manuscript further.

- Again, the authors believe that “A low-dose AdipoRon treatment may elicit similar pro-neurogenic effect by up-regulating the BDNF pathway via AMPK activation.” (line 235), do they have any evidence of AMPK activation in this case? The idea that it could be AMPK over activation could inhibit BDNF is interesting but should be supported by experimental evidences.

Author Response

We would like to thank your active participation in the review process. We have greatly improved the quality of the manuscript in this revision. We hope that the quality is beyond satisfactory. Please see the point-to-point responses below:

The paper by Lee and colleagues is interesting, but I feel as though further experimental evidences are lacking in order to make the results more significant and relevant for the field. Indeed, it would be interesting to include data correlating the observed evidences with AMPK activation, BDNF direct increase or its effects on synaptogenesis.

Moreover, I have some suggestions I feel the authors should take into consideration before the manuscript can be accepted for publication:

- I would adjust the title as I don’t believe the part stating “but high dose shows opposite effects” fully depicts the meaning of the paper (maybe something along the lines of “AdipoRon treatment induces a dose-dependent response in adult hippocampal neurogenesis”)

Thank you very much for your suggestion. We have revised our title as AdipoRon treatment induces a dose-dependent response in adult hippocampal neurogenesis.

- The authors need to address some English mistakes, for example:

         - Abstract: line 26 “yet to be examined”; line 28 “either 20 mg/kg (low dose) or 50 mg/kg (high dose)..”; line 37 “Conversely” is not correct as the 50 mg/kg dose shows the opposite effect, not a concordant one (so maybe “on the contrary?”): line 39 “the results suggest…”

         - Introduction: line 46 “secreted by the adipose tissue”

         - Results, line 124 I would rephrase as “highlighting differential effects of …”

         - Discussion, line 199 “the findings echoed previous reports..”; The sentence in line 199-202 is not clear.;

line 208 “BDNF is a neurotrophic factor THAT promotes adult neurogenesis”

            Thank you so much for pointing out the mistakes. We have revised and corrected the sentences accordingly.

- In the introduction, it is not clear why adiponectin’s anti-diabetic effects are mentioned right away when discussing its effect on the brain, maybe they should be commented on when discussing the action of adipoRon in diabetes (e.g. line 69).

Thank you so much for your suggestion, we have greatly improved the readability of introduction, Line 75-88.

“Circulating adiponectin levels are decreased in individuals with obesity and Type 2 diabetes (28). Adiponectin possesses antidiabetic property (29-31) by promoting fatty acid oxidation through AdipoR1/AMPK or AdipoR2/ PPAR-α pathway (32). The antidiabetic effect of adiponectin can be mimicked by administering a small, orally active, adiponectin receptor agonist, AdipoRon (33). Mechanistically, AdipoRon binds to both AdipoR1 and AdipoR2 in vitro with a greater affinity for AdipoR1 (33). Functionally, the antidiabetic effect of AdipoRon is completely abolished under the condition of double knockout of AdipoR1 and AdipoR2, but not single knockout of either adiponectin receptors (33). Further investigations reveal its anti-inflammatory (34), anti-atherosclerotic (35), hepatoprotective (36), and cardioprotective (37) properties in animal models. AdipoRon can be detected in the brain upon systemic injection (38) and oral administration (8), implicating its ability to cross the blood-brain barrier (BBB). Former studies focused on examining the pharmacological effect of AdipoRon in neurocognitive diseases (8, 38), its neurotrophic effect in physiological condition has yet to be examined.”

Moreover, if there is more than 1 paper discussing the neuroprotective effects of AdipoRon (other than ref 20), they should be included and commented upon

- In the results, when describing Figure 1 A-C, it should be stated that the results refer to both dosages (none of them affects travelling distance/speed)

            We have specified both dosages in Section 2.1, Line 101 and 105.

- In Figure 1 (and in the subsequent figures), if the number indicated in the column (7) refers to the number of animals analyzed, this should be stated in the legend. Moreover, it should be stated here if the data refers to the mean +/- SD or SEM.

We have elaborated the number of animal used and Mean+/- SEM in the corresponding figure legends.

- In figure 1 the legend is present only in Figure 1C. I understand that this is the same for the adjacent panels, but it is misleading that there is no clear labeling in Panel 1A and 1B, I would thus either add the legend on the X axis or anyways add it to make it clearer. The same applies for Figure 5

            We have removed the legends in the bar graphs.

- I think Figure 1D1 is a bit redundant to Figure 1D2 and confusing, I suggest keeping only Figure 1D2 which gives a clearer idea of the result obtained

We have removed Figure 1D1. And D2 became Figure 1F

- In figure 2, how many images were stained and quantified per condition? This should be stated in the legend (I see it is present in the methods but it would be useful if it were reported here too).

We have specified 8 sections were counted per animal, 5 animals per treatment group.

Moreover, scale bars are present only in the magnification of figure 2 and in Figure 2D, they should be present in all images (This is also true for Figure 3 and Figure 4)

             We have added back the scale bars to the corresponding figures.

- In line 153, the authors state that “20 mg/kg AdipoRon, but not 50 mg/kg AdipoRon treatment, significantly suppressed neuronal differentiation” but in Figure 4E the opposite seems to be true (The 50 dosage suppresses neuronal differentiation).

We have amended the line as in 50 mg/kg AdipoRon, but not 20 mg/kg AdipoRon treatment, significantly suppressed neuronal differentiation.

- Figures 4F-4H should be commented upon in the text. Moreover, it is not clear to what condition these imaged refer to: The vehicle, the 20 dosage or the 50 dosage. Representative images of the staining for each condition should be reported.

We have presented the representative images for vehicle, 20, and 50 AdipoRon treatments for BrdU/DCX co-labelled cells.

- The authors should elaborate on how they believe that “The increased adiponectin and BDNF levels may contribute to the indirect action of AdipoRon on adult neurogenesis.” (line 209)

            We have clarified in Discussion, Line 213-233

“Cell quantification showed that a low-dose (20 mg/kg) AdipoRon treatment pro-moted hippocampal cell proliferation with increased serum levels of adiponectin and BDNF. With the ability to cross the blood-brain barrier (5), adiponectin may mediate the AdipoRon action on hippocampal cell proliferation. The direct action of adiponectin on hippocampal cell proliferation is evidenced in vivo (14) and in vitro (5, 43). Be-sides, BDNF mediates hippocampal cell proliferation through TrkB (44). Circulating and brain BDNF levels are positively correlated in humans and rodents (45). The elevated serum BDNF level echoes our latest finding, illustrating that a low-dose Adipo-Ron treatment raises BDNF level in the dentate subregion (Lee et al., unpublished data). While the action of BDNF encompasses all stages of adult neurogenesis (46), we did not observe any neurotrophic effects of AdipoRon on neuronal differentiation and survival at a low dose. We speculate that BDNF could play a more critical role in forming dendritic arbours and spines (47). A recent study shows that AdipoRon administration promotes spine formation in the apical dendrites of CA1 neurons in both 5×FAD and 5×FAD; Adipo-knockout mice (8), while we also observed an increased spine density in DG granule neurons in mice receiving low-dose AdipoRon treatment (Lee et al., unpublished data). The ability to cross BBB facilitates the direct action of AdipoRon on the hippocampus (8, 38), where adiponectin receptors are expressed (12). Intracerebral ventricular infusion of AdipoRon promotes cell proliferation in DG of APP/PS1 mice, and AdipoRon also restores Aβ-impaired neural stem cell proliferation (48). Collectively, this study presents the potential direct and indirect actions of AdipoRon on hippocampal cell proliferation.”

- Do the authors have any evidence of hippocampal expression of BDNF? I see they state that evidences demonstrate a positive correlation between circulating and brain bdnf levels (208), but the actual BDNF concentration in the brain could be useful.

Yes, we observed an elevation of BDNF level in the dentate gyrus subregion in a mouse cohort receiving 20 mg/kg AdipoRon. Discussion, Line 220-221.

“…, illustrating that a low-dose AdipoRon treatment raises BDNF level in the dentate sub-region (Lee et al., unpublished data).”

- The authors state that “BDNF could play a more critical role in dendritic morphogenesis (27). It will be of interest to examine dendritic structures of newborn neurons in future studies.” (213-214). Do they have any evidence of this, such as immunohistochemical staining or gene expression analysis? I believe this would enrich the manuscript further.

We apologise we did not perform further IHC or gene expression analyses. Ng et al. have reported that AdipoRon administration promotes dendritic spine formation in the CA1 pyramidal cells of the 5xFAD AD mouse model. However, the level of BDNF in the hippocampus is not examined. We have a similar observation in mice receiving the same dosage of AdipoRon with increased spine formation in the DG granule cells and increased BDNF levels in the dentate gyrus. Discussion, Line 225-228:

“A recent study shows that AdipoRon administration promotes spine formation in the apical dendrites of CA1 neurons in both 5×FAD and 5×FAD; Adipo-knockout mice (8), while we also observed an increased spine density in DG granule neurons in mice receiving low-dose AdipoRon treatment (Lee et al., unpublished data).”

- Again, the authors believe that “A low-dose AdipoRon treatment may elicit similar pro-neurogenic effect by up-regulating the BDNF pathway via AMPK activation.” (line 235), do they have any evidence of AMPK activation in this case?

In another separate study, we observed AMPK activation in the dentate gyrus. Discussion, Line 278-280:

“Likewise, low-dose AdipoRon treatment increased AMPKThr172 phosphorylation in the dentate subregion (Lee et al., unpublished data).”

The idea that it could be AMPK over activation could inhibit BDNF is interesting but should be supported by experimental evidences.

Thank you so much for the idea. We have further elaborated this possibility in Discussion, Line 272-275:

“Overactivation of AMPK by AICAR infusion or AMPK overexpression induces AMPKThr172 phosphorylation but reduces BDNF expression in the hippocampal CA1 (62). Memory deficit is subsequently presented in contextual fear conditioning after AMPK activation (62).”

Round 2

Reviewer 1 Report

The authors answered my concerns. 

Author Response

Once again, we thank you for the time you put in reviewing our paper. Your inputs have been precious in the eventuality of a publication.

Reviewer 2 Report

The manuscript is much improved and I thank the authors for this. I still have one minor comment which is that in Figure 4, panels I, J and K only refer to a zoom in and merge of the vehicle. I suggest either including the same for the 20 and 50 dosage or remove it altogether. Thank you for your work

Author Response

We have removed Figure 4, panels I, J and K accordingly.

Once again, we thank you for the time you put in reviewing our paper and look forward to meeting your expectations. Your inputs have been precious in the eventuality of a publication.